# Fractional Flow Reserve Cardio-Oncology Effects on Inpatient Mortality, Length of Stay, and Cost Based on Malignancy Type: Machine Learning Supported Nationally Representative Case-Control Study of 30 Million Hospitalizations

**DOI:** 10.3390/medicina58070859

**Published:** 2022-06-28

**Authors:** Siddharth Chauhan, Dominique J. Monlezun, Jin wan Kim, Harsh Goel, Alex Hanna, Kenneth Hoang, Nicolas Palaskas, Juan Lopez-Mattei, Saamir Hassan, Peter Kim, Mehmet Cilingiroglu, Konstantinos Marmagkiolis, Cezar A. Iliescu

**Affiliations:** 1Department of Cardiology, University of Texas Health Science Center at Houston, Houston, TX 77030, USA; jin.wan.kim@uth.tmc.edu (J.w.K.); harsh.goel@uth.tmc.edu (H.G.); alexander.b.hanna@uth.tmc.edu (A.H.); kennethhoang@uabmc.edu (K.H.); ciliescu@mdanderson.org (C.A.I.); 2Department of Cardiology, The University of Texas MD Anderson Cancer Center, Houston, TX 77030, USA; dominique.monlezun@gmail.com (D.J.M.); nlpalaskas@mdanderson.org (N.P.); juan.lopezmattei@leehealth.org (J.L.-M.); sahassan1@mdanderson.org (S.H.); pkim123@gmail.com (P.K.); cilingiroglumehmet@gmail.com (M.C.); c.marmagiolis@gmail.com (K.M.)

**Keywords:** fractional flow reserve, percutaneous coronary intervention, cancer, malignancy, cardio-oncology, rectal cancer, Hodgkin’s lymphoma

## Abstract

*Background and Objectives:* There are no nationally representative studies of mortality and cost effectiveness for fractional flow reserve (FFR) guided percutaneous coronary interventions (PCI) in patients with cancer. Our study aims to show how this patient population may benefit from FFR-guided PCI. *Materials and Methods*: Propensity score matched analysis and backward propagation neural network machine learning supported multivariable regression was performed for inpatient mortality in this case-control study of the 2016 National Inpatient Sample (NIS). Regression results were adjusted for age, race, income, geographic region, metastases, mortality risk, and the likelihood of undergoing FFR versus non-FFR PCI. All analyses were adjusted for the complex survey design to produce nationally representative estimates. *Results*: Of the 30,195,722 hospitalized patients meeting criteria, 3.37% of the PCIs performed included FFR. In propensity score adjusted multivariable regression, FFR versus non-FFR PCI significantly reduced inpatient mortality (OR 0.47, 95%CI 0.35–0.63; *p* < 0.001) and length of stay (LOS) (in days; beta −0.23, 95%CI −0.37–−0.09; *p* = 0.001) while increasing cost (in USD; beta $5708.63, 95%CI, 3042.70–8374.57; *p* < 0.001), without significantly increasing complications overall. FFR versus non-FFR PCI did not specifically change cancer patients’ inpatient mortality, LOS, or cost. However, FFR versus non-FFR PCI significantly increased inpatient mortality for Hodgkin’s lymphoma (OR 52.48, 95%CI 7.16–384.53; *p* < 0.001) and rectal cancer (OR 24.38, 95%CI 2.24–265.73; *p* = 0.009). *Conclusions*: FFR-guided PCI may be safely utilized in patients with cancer as it does not significantly increase inpatient mortality, complications, and LOS. These findings support the need for an increased utilization of FFR-guided PCI and further studies to evaluate its long-term impact.

## 1. Introduction

Percutaneous coronary intervention (PCI) is the mainstay of cardiologist interventions in acute coronary events [1]. However, there are opposing viewpoints on whether non-acute patients warrant coronary interventions [2,3]. By identifying obstructive lesions and reducing the number of deployed stents, Fractional flow reserve (FFR)-guided PCI, has been shown to improve outcomes when utilized in the general population. However, its benefit and safety in patients with cancer have not been measured.

Three large trials, FAME, FAME2, and FAME3 have shown that FFR-guided PCI leads to favorable outcomes in comparison to PCI alone and is non-inferior to CABG [4,5,6]. The DEFER study was the first to show that patients who deferred intervention based on FFR had no significant difference in outcomes compared to medical therapy [7]. However, the external validity of these studies is limited when applied to patients with cancer, as this group was not included. No studies in the literature have directly compared FFR vs. non-FFR PCI outcomes in this patient population. As FFR requires more instrumentation with associated risk of complications and cost, our study attempted to measure the immediate benefits and risks of FFR in hospitalized patients with cancer.

## 2. Materials and Methods

### 2.1. Data Source

The 2016 National Inpatient Sample (NIS) was chosen for this study’s data set as it utilizes ICD-10 codes and is the latest available, allowing generalizable results to current clinical practice. The NIS began in 2004 as a data collection from select hospitals and expanded in 2012 to encompass discharge data from all HCUP participating hospitals. In 2016, the NIS data coding adopted the International Classification of Diseases, Tenth Revision, Clinical Modification (ICD-10-CM). The NIS currently accounts for approximately 1 in 5 discharges from all community hospitals in the United States. The NIS is sponsored by United States’ Agency for Healthcare Research and Quality under the Department of Health and Human Services (HHS) and maintained within the Healthcare Cost and Utilization Project (HCUP).

### 2.2. Study Design

This is the first known nationally representative multi-center analysis of inpatient mortality and total costs among all eligible hospitalized adults by PCI (with vs. without FFR) and cancer (yes/no, including overall and comparatively by primary organ site). The 2016 NIS dataset was selected for this study because it is among the most recent, was the first to use ICD-10 codes, and more closely reflects current clinical trends in PCI use versus earlier years. Study inclusion criteria were all 2016 NIS hospitalizations for adults 18 years or older during 2018. The NIS is classified as a limited data set by the United States’ Agency for Healthcare Research and Quality under the Department of Health and Human Services. As an HCUP limited data set, the NIS does not require institutional review board (IRB) approval under HIPAA. The study was performed under the ethical principles in the 1975 Declaration of Helsinki and related global bioethical standards

Subjects undergoing PCI were identified by the ICD-10 procedure codes of 00.66 (percutaneous transluminal coronary angioplasty), 36.06 (insertion of non-drug-eluting coronary artery stent(s)), or 36.07 (insertion of drug-eluting coronary artery stent(s)). ICD-10 codes were also used to identify variables such as cardiac arrest, demographics, comorbidities, and outcomes. HCUP tools such as the Clinical Classification Software, which had been used prior to the NIS 2016 dataset for such purposes as classifying cancer (e.g., by primary type, current versus historical), were not used in this study because they were found to be unreliable when applied to the 2016 dataset’s ICD-10 data.

### 2.3. Descriptive and Bivariable Statistical Analysis

Descriptive statistics for demographics (i.e., age, sex, race, insurance) and comorbidities were performed for the full sample. Comorbidities were selected for analysis (and identified in the dataset by their ICD-10 scores) on the basis of their clinical and/or statistical significance for similar studies in the existing literature. The comorbidities included in this study were diabetes, hypertension, peripheral vascular disease, hyperlipidemia, smoking, obesity, poor diet, stroke, congestive heart failure, cardiac arrest, myocardial infarction, cardiogenic shock, valvular disease, HIV, alcohol abuse, opioid abuse, anemia, chronic obstructive pulmonary disease, coagulopathy, depression, cirrhosis, chronic kidney disease, and malignancy (overall and by primary malignancy type).

Bivariable analysis was conducted according to inpatient mortality (yes/no). For continuous variables, independent sample t tests were performed to compare means and Wilcoxon rank sum tests were performed for medians. For categorical variables, Pearson chi square tests or Fisher exact tests were performed to compare proportions.

### 2.4. Regression Statistical Analysis, Machine Learning Analysis, and Model Optimization

Mortality (yes/no) was measured as the primary outcome. Secondary outcomes included length of stay (LOS in days), cost (in United States dollars), and complication types (including post-procedure bleeding, stroke, and acute kidney injury). Secondary predictors included PCI with single vessel, multivessel, drug eluting stent, and bare metal stents. The regression models were optimized according to the following sequential process in order to maximize validity (internal and external) and replicability. First, clinically or statistically significant variables were identified in the existing literature, clinical practice, and bivariable analysis for consideration in the final regression models. Second, forward and backward stepwise regression was done on these variables to augment decision-making on the variables ultimately included in the final regression models. Third, backward propagation neural network machine learning was used to generate regression results for comparison of accuracy and root mean squared error. Fourth, additional assessment of regression results was done via correlation matrix, area under the curve, Akaike and Schwarz Bayesian information criterion, Hosmer–Lemeshow goodness-of-fit test, variance inflation factor and tolerance, multicollinearity, and specification error. Fifth, final models and variables were iteratively run with fine tuning until the above process reached and confirmed optimal performance. All models were adjusted for age, race, sex, income, region, urban density, metastases, and acute coronary syndrome (ACS) mortality risk as calculated by the NIS according to diagnosis-related group (DRG). Other variables, as determined by machine learning analysis and diagnostic testing, were excluded to produce the most statistically and clinically justifiable models.

### 2.5. Machine Learning-Augmented Propensity Score Adjusted Multivariable Regression (ML-PSr)

ML-PSr was used for the above regression models [8,9,10,11,12] to generate a propensity score for the likelihood of undergoing inpatient PCI with versus without FFR (the treatment, utilizing the same above variables used in the final regression model given the double propensity score adjustment method). Balance was confirmed among blocks, and the propensity score was included as an adjusted variable in the final regression models. We selected a causal inference approach (propensity score adjustment) as it is a widely accepted methodology of reducing, but not eliminating, the effects of confounding variables and selection bias. Other causal inference approaches such as fixed, random, and mixed effects have the advantage of reducing unobserved variable bias but were not optimal for this study as the NIS lacked adequate repeat hospitalizations from the same subjects. Propensity score adjustment is superior to covariate adjustment without the propensity score, as it can produce a more complex propensity score model (i.e., can analyze interactions and higher order terms to yield the best estimated probability of treatment assignment) without the risks of over-parameterizing. Diagnostic analysis of the final models can also be done with this approach to confirm superior performance over simple covariate adjustment without the propensity score. Lastly, propensity score adjustment was chosen over competing propensity score techniques due to its superior performance in the appropriate context (according to latest statistical theory and adequate diagnostic quantitative testing of final models in cardiovascular studies [13,14] and because inclusion in the final regression models confirmed sufficient performance by the above specified optimization process. Propensity score matching (PSM) was additionally conducted to estimate the average treatment effect (ATE) for PCI with versus without FFR using the same variables identified in the final above model for ML-PSr for mortality to compare to the post-regression marginal effect, given the familiarity, popularity, and ease of interpretation of this technique for clinical audiences and to allow more robust analysis across diverse techniques of the possible association between mortality and FFR.

The hybrid analytic approach above integrates a traditional statistical method of frequentist-based multivariable regression (supported by propensity score-based causal inference analysis) and supervised learning-based machine learning and has several advantages. Causal inference results are more familiar to medical audiences and can be confirmed and replicated automatically by machine learning, which yields more rapid and accurate results compared with traditional statistics. This accelerates real-time findings for large high-dimensional datasets as is already done for economic sectors outside the medical sciences.

### 2.6. Stratification and Sub-Group Analysis

Regression models were separately stratified by cancer status (present/absent), active versus prior cancer, and metastatic versus non-metastatic malignancy.

### 2.7. Model Validation, Reporting, and Analytic Software

An academic physician-data scientist and biostatistician (DJM) confirmed that the final regression models were sufficiently supported by the existing literature and clinical and statistical theory. Mean values are reported with standard deviation (SDs). Fully adjusted regression results were reported with 95% confidence intervals (CIs) with statistical significance set at a 2-tailed *p*-value of <0.05. Statistical analysis was performed with STATA 17.0 (STATACorp, College Station, TX, USA), and machine learning analysis was performed with Java 9 (Oracle, Redwood Chores, CA, USA).

## 3. Results

### 3.1. Descriptive Statistics

Of the 30,195,722 adult hospitalized patients nationally in 2016, 1156,349 (3.83%) underwent PCI with 39,150 (3.39%) of these being performed with FFR and 2.78% dying inpatient (Table 1). Patients undergoing PCI with versus without FFR were significantly less likely to be non-white (24.58% versus 27.85%), be in the lowest income quartile (29.66% versus 31.22%), have any complications (4.06% versus 5.18%) die inpatient (1.10% versus 2.84%), have a lower mean length of stay in days (4.70 [SD 5.49] versus 5.51 [7.21]) and lower total cost (−$5478.80 [SD −33,782]) (all *p* < 0.01). Of patients receiving FFR-guided PCI, 11.14% were conducted in patients with cancer, and 7.84% following inpatient cardiac stress tests. The most common primary malignancies in which FFR-guided PCI was performed included prostate (21.45%), skin (15.34%), breast (14.53%), lung (8.30%), and bladder (7.27%). In sub-group analysis among the 3814 patients with an inpatient cardiac stress test followed by PCI, 255 (6.69%) had FFR- guided PCI.

### 3.2. Multivariable Regression

In propensity score adjusted multivariable regression, FFR versus non-FFR PCI significantly reduced inpatient mortality (OR 0.47, 95%CI 0.35–0.63; *p* < 0.001) and LOS (in days; beta −0.23, 95%CI −0.37–−0.09; *p* = 0.001) for patients overall while increasing cost (in USD; beta $5708.63, 95%CI 3042.70–8374.57; *p* < 0.001), and without significantly increasing complications overall (Table 2). Post-regression marginal effect indicated that for patients overall, FFR versus non-FFR PCI significantly reduced mortality by 1.33% (95%CI 1.01–1.65, *p* < 0.001), comparable with results from PSM.

FFR versus non-FFR PCI was not significantly associated with increase in patients with cancer mortality, LOS, cost, or complication types (including post-procedure bleeding, stroke, or acute kidney injury) (Table 3). In regression analysis stratified by cancer status, FFR versus non-FFR PCI significantly reduced mortality (OR 0.47, 95%CI 0.35–0.63; *p* < 0.001) for patients without cancer and non-significantly reduced mortality for patients with cancer (OR 0.56, 95%CI 0.29–1.07; *p* = 0.078).

This was similar to patients both with and without stress tests. In regression analysis stratified by active versus prior cancer status, FFR versus non-FFR PCI non-significantly reduced mortality for prior patients with cancer (OR 0.56, 95%CI 0.25–1.25; *p* = 0.16) and patients with active malignancy (OR 0.55, 95%CI 0.18–1.71; *p* = 0.30). Amongst all primary malignancies analyzed for FFR versus non-FFR PCI, mortality was significantly increased for Hodgkin’s lymphoma (OR 52.48, 95%CI 7.16–384.53; *p* < 0.001) and rectal cancer (OR 24.38, 95%CI 2.24–265.73; *p* = 0.009).

Among patients with active cancer, there was no significant association of FFR versus non-FFR PCI and complications: overall, cardiogenic shock, cardiac arrest, hemorrhage, pericardial effusion, vessel puncture, non-puncture vascular complication, stroke, line associated blood stream infection, pulmonary embolus, or acute kidney injury. Neither was there a significant association between FFR versus non-FFR PCI and mortality (or the above complications) when the PCI (separately) involved single vessel, multivessel, drug eluting stent, or bare metal stents. Among patients with active metastatic disease, there was no significant association of FFR versus non-FFR PCI and mortality nor the above complications.

## 4. Discussion

Patients with cancer who undergo PCI have been found to have improved inpatient mortality [9]. However, there are reports of increased complications, including 90-day admission, and bleeding events that abbreviate DAPT therapy which can interrupt required cancer treatment procedures [15,16]. These findings support the idea of reducing unnecessary stenting through iFR/FFR guidance in cancer patients, but how clinical outcomes may be adversely affected by additional instrumentation is unknown. Therefore, our study sought to show that FFR is safe in the inpatient setting for patients with cancer by demonstrating that significant differences were not present according to inpatient mortality, LOS, cost or complication types in FFR vs. non-FFR PCI. Further studies may be done to measure long term mortality following FFR-guided PCI. Our findings help contribute to the mounting literature that supports the use of established cardiac medical interventions for patients with cancer.

Translating general cardiac interventions for use in the cancer population is important as we uncover the mechanisms of how cardiovascular disease (CVD) and cancer interact [4,17,18]. Increased proinflammatory states and hypercoagulability of malignancy contribute to higher rates of atherosclerosis. Chemotherapeutics also expose patients with cancer to unpredictable cardiotoxic side effects [19]. Conversely, many types of common cancers, including lung, bladder and colon are observed to occur at a higher incidence rate in patients with known coronary artery disease compared with the general population [20]. Higher CVD burden is further illustrated by one large cohort study of nearly 13,000 patients who underwent PCI, of which adult cancer survivors comprised a notably high proportion of one in every thirteen patients [21]. With the increasing evidence of high cardiovascular risk, refining the role of PCI and FFR in patients with cancer is crucial.

PCI in patients with cancer carries increased risks of hemorrhage, hematoma and vascular perforation in the setting of malignancy [22]. Physicians may consider that there is a potential increased risk arising from the extended instrumentation of FFR in this population and during long term follow up. Randomized controlled trials have demonstrated significantly better outcomes for cardiac morbidity and mortality with FFR PCI. However, before these are measured in patients with cancer, immediate safety and inpatient mortality should be shown to be acceptable as we have done in our study.

Our study did not find statistically significant differences in inpatient mortality, LOS, cost or complication types (including post-procedural bleeding, stroke, or acute kidney injury) for patients with cancer undergoing FFR-guided PCI. Immediate decreases in inpatient mortality from FFR may not be evident based on current hypotheses behind how FFR confers its mortality benefit. The FAME trial suggests that by reducing stent placement within adequate FFR lesions, the reduced rates of in-stent thrombosis/restenosis over time grants improvements in one-year mortality [4]. Further studies will need to be done to measure differences in follow up mortality. However, as FFR is not associated with increased immediate morbidity or mortality in cancer patients, it is reasonable to utilize FFR on a case-by-case basis. For cancer patients, whose treatments are complicated by impending surgeries, chronic bleeding risk, thrombotic risk and complicated medication regimens, flexibility in the decision to stent and sustain antiplatelet medication may provide a yet undefined benefit to be explored in the future.

Another significant finding from our study pertains to patients with Hodgkin’s lymphoma and rectal cancer, in whom FFR-guided PCI showed increased inpatient mortality compared to PCI. A precise mechanism is not known, but we can consider unique treatment strategies and complications of these cancers which may provide some explanation. Mediastinal radiotherapy and cardiotoxic anthracycline-based chemotherapeutic regimens unique to Hodgkin’s lymphoma have been proposed as significant risk factors for the development and progression of CAD through endothelial damage to coronary arteries [23,24,25]. In colorectal cancer, 5-fluorouracil (5-FU)-based chemotherapeutic regimens have been proposed to be an under-recognized and under-appreciated cause of ischemic cardiotoxicity via vascular endothelial dysfunction, microthrombi formation, impaired erythrocyte oxygen delivery, and coronary vasospasm [26,27]. Perhaps another important consideration is that gastrointestinal bleeding in colorectal cancer may further exacerbate cardiac ischemia even in the setting of non-hemodynamically significant coronary artery stenoses via supply-demand mismatch.

Given the statistical significance of these two cancers and FFR associated inpatient mortality, it is possible that they may have cardiovascular phenotypes that do not benefit from an FFR-guided PCI approach. The wide confidence intervals limit our ability to draw firm conclusions for cancers such as Hodgkin’s lymphoma and colorectal cancer, likely secondary to smaller sample sizes. Larger studies are needed to further validate these findings.

## 5. Conclusions

FFR vs. non-FFR PCI is associated with significantly reduced inpatient mortality and LOS without increases in post-procedure complications. With stratification by cancer status, FFR vs. non-FFR PCI does not show significant differences in inpatient mortality, LOS, cost or complication types in patients with cancers. Interestingly, FFR vs. non-FFR PCI is associated with higher inpatient mortality in Hodgkin’s lymphoma and rectal cancer, however, definitive conclusions are limited by wide confidence intervals and small sample sizes. As FFR is not associated with increased inpatient adverse events nor mortality, further studies should be aimed towards differences in follow up mortality. In the present, for cancer patients whose medical care is complicated by risks of both bleeding and thrombotic events, these findings support physicians’ decisions to utilize FFR according to the unique history of each patient.

There are several limitations to mention regarding our study. Firstly, our study focused on inpatient mortality, while the out-of-hospital benefits and complications of FFR-guided PCI were not evaluated. Longer term studies are required to evaluate the benefits of FFR-guided PCI in patients with cancer. Second, the relatively small proportion of patients with cancer undergoing FFR-guided PCI may have limited the study’s power to detect improved outcomes. This further reflects how FFR is underutilized in patients with cancer, limiting the amount of data available to assess its efficacy. Finally, as a case-control study, definitive cause–effect conclusions may not be drawn, particularly with respect to an associated increased mortality noted with FFR-guided PCI in Hodgkin’s lymphoma and rectal cancer patients.

## Figures and Tables

**Table 1 medicina-58-00859-t001:** Demographic statistics and bivariable analysis by PCI with versus without FFR (*n* = 1,156,349).

Variable, (%)	Sample	PCI Alive (97.22%)		*p*-Value
		Without FFR (96.91%)	With FFR (3.39%)	
*Demographic*				
Age, mean (SD)	64.80 (12.99)	64.77 (13.04)	65.50 (11.63)	<0.001
Female	38.24	38.25	37.81	0.429
Non-white race	27.75	27.85	24.58	<0.001
*Insurance*				0.004
Commercial	26.87	26.90	26.20	
Medicare	55.37	55.30	57.35	
Medicaid	10.64	10.66	9.94	
Veterans Affairs	2.87	2.88	2.73	
None	4.25	4.26	3.78	
*Income quartile*				0.006
1st (lowest)	31.17	31.22	29.66	
2nd	26.60	26.60	26.64	
3rd	23.62	23.57	25.02	
4th (highest)	18.61	18.61	18.69	
*Comorbidities*				
Diabetes	29.17	29.11	30.97	<0.001
Hypertension	80.83	80.66	85.82	<0.001
Peripheral vascular disease	7.83	7.68	7.83	0.623
Hyperlipidemia	67.09	66.89	72.91	<0.001
Smoking	2.06	2.07	1.79	0.081
Obesity	20.37	20.33	21.44	0.017
Poor diet	0.30	0.30	0.24	0.363
Stroke	4.46	4.46	4.52	0.785
Heart failure	31.19	31.31	27.78	<0.001
Cardiac arrest	4.22	4.28	2.55	<0.001
Valvular disease	17.48	17.56	15.42	<0.001
Smoking	2.06	2.07	1.79	0.081
HIV	0.19	0.19	0.14	0.285
Alcoholism	3.95	3.96	3.51	0.045
Opiate dependence	0.81	0.81	0.79	0.851
Anemia	17.95	18.01	16.37	<0.001
COPD	19.38	19.35	20.15	0.078
Coagulopathy	7.41	7.46	5.98	<0.001
Depression	9.34	9.31	10.01	0.037
Cirrhosis	1.10	1.10	0.89	0.080
Chronic kidney disease (3–5)	16.29	16.33	15.34	0.020
*Acute myocardial infarction*	48.31	48.65	38.49	<0.001
STEMI	15.05	15.37	5.98	<0.001
NSTEMI/UA	33.53	33.56	32.75	0.135
Cardiogenic shock	4.95	5.06	1.98	<0.001
*Cancer*	11.06	11.06	11.14	0.822
Active	2.64	2.66	2.13	0.005
Metastasis	0.72	0.73	0.29	<0.001
*Inpatient*				
Mortality risk, mean (SD)	0.72 (0.99)	0.72 (0.99)	0.64 (0.92)	<0.001
Mortality	2.78	2.84	1.10	<0.001
LOS, mean (SD)	5.48 (7.16)	5.51 (7.21)	4.70 (5.49)	<0.001
Cost USD, mean (SD)	108,347.90 (133,058.10)	108,533.10 (134,056.10)	103,054.30 (100,274.10)	<0.001
*Complications*	5.14	5.18	4.06	<0.001
Bleed	0.94	0.94	0.80	0.218
Stroke	0.14	0.14	0.08	0.125
Acute kidney injury	0.11	0.11	0.11	0.947

**Table 2 medicina-58-00859-t002:** Propensity score adjusted machine learning supported multivariable regression of mortality by FFR versus non-FFR PCI.

Variable	OR (95.0% CI)	*p*-Value
Age	1.09, 1.09–1.10	<0.001
Female	1.07, 0.98–1.17	0.129
Race, nonwhite	1.02, 0.95–1.09	0.652
*Income quartile*		
1st (lowest)	Reference	
2nd	1.32, 1.23–1.42	<0.001
3rd	1.52, 1.40–1.65	<0.001
4th (highest)	2.04, 1.86–2.23	<0.001
*Region*		
New England	Reference	
Mid Atlantic	1.32, 1.13–1.56	0.001
East North Central	1.61, 1.37–1.88	<0.001
West North Central	1.82, 1.52–2.18	<0.001
South Atlantic	1.84, 1.57–2.15	<0.001
East South Central	1.80, 1.51–2.15	<0.001
West South Central	2.48, 2.11–2.92	<0.001
Mountain	1.74, 1.44–2.09	<0.001
Pacific	2.26, 1.92–2.66	<0.001
*Urban density*		
>=1 million central	Reference	
>=1 million fringe	0.91, 0.84–0.99	0.025
250,000–999,999	1.03, 0.95–1.11	0.509
50,000–249,999	1.03, 0.92–1.14	0.639
Micro	1.03, 0.92–1.14	0.633
<Micro	0.92, 0.82–1.40	0.189
Acute coronary syndrome	1.26, 1.19–1.34	<0.001
FFR	0.47, 0.37–0.61	<0.001
*Cancer*		
*Cancer*	0.90, 0.82–0.98	0.013
With FFR	1.20, 0.63–2.29	0.580
Metastasis	1.91, 1.56–2.33	<0.001
Mortality risk	1.04, 1.01–1.07	0.011

**Table 3 medicina-58-00859-t003:** Propensity score adjusted machine learning supported multivariable regression of mortality by FFR versus non-FFR PCI among the top 5 cancers for FFR.

Cancer	OR (95.0%CI)	*p*-Value
Overall	1.15, 0.58–2.30	0.686
*Primary malignancy*		
Prostate	1.80, 0.59–5.55	0.304
Skin	1.16, 0.23–5.78	0.858
Breast	0.67, 0.08–5.86	0.720
Lung	1.27, 0.27–6.09	0.764
Bladder	1.28, 0.14–11.39	0.822

## Data Availability

Not applicable.

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
