# Peer review of "Fractional Flow Reserve Cardio-Oncology Effects on Inpatient Mortality, Length of Stay, and Cost Based on Malignancy Type: Machine Learning Supported Nationally Representative Case-Control Study of 30 Million Hospitalizations"

_medicina, 2022, doi:10.3390/medicina58070859_

Round 1

Reviewer 1 Report

Authors investigated fractional flow reserve (FFR) guided percutaneous coronary interventions (PCI) in patients with cancers is associated with significantly reduced inpatient mortality and length of stay (LOS) without increases in post-procedure complications. The study strategy is notable, and the results can indicate the conclusion. However, the reviewer has some concerns as follows. Please consider and reply.

1.     Tables do not work to understand the results. Authors should add the table of the propensity score adjusted multivariable regression for FFR guided PCI vs non-FFR guided PCI (Participants characteristics and results). In addition, Authors should declare details of the comorbidities (Diabetes, hypertension, peripheral vascular disease, hyperlipidemia, smoking, obesity, poor diet, stroke, congestive heart failure, cardiac arrest, myocardial infarction, cardiogenic shock, valvular disease, HIV, alcohol abuse, opioid abuse, anemia, chronic obstructive pulmonary disease, coagulopathy, depression, cirrhosis, chronic kidney disease, and malignancy) and complications (post procedure bleeding, stroke, or acute kidney injury).

2.     Authors’ study subject is unclear. Please explain more about the background in patients with cancer who underwent FFR guided PCI. The reviewer thought authors should show the reference that PCI has a risk of complications in patients with cancers. For example, Kwok CS, et al. reported that within 90-day readmission for AMI after PCI was higher in patients with active cancer [DOI: 10.1093/eurheartj/ehaa1032]. Then, FFR guided PCI will have more significance if authors can explain about a risk of complications during PCI in patients with cancers.

3.     Authors should declare which factors were propensity score matched.

4.     Authors should declare the cardiovascular mortality in addition to all-cause mortality.

5.     Please declare the mortality risk in Table 1 in methods.

Reviewer 2 Report

‘FFR Cardio-Oncology Effects on Inpatient Mortality, Length of Stay, and Cost Based on Malignancy Type: Machine Learning Supported Nationally Representative Case-Control Study of 30 Million Hospitalizations’

Since this Manuscript [MS] presents an analysis of a large Nationally Representative Case-Control Study [of 30 Million Hospitalizations], its results could provide physicians with important data RE: FFR-guided PCI, as a generally safe procedure that can be utilized in many patients with cancer and CVD.

In addition, highlighting a significant study finding that FFR vs non-FFR PCI is associated with higher inpatient mortality in cases of Hodgkin’s lymphoma and rectal cancer is valuable [pending further studies with larger sample sizes to elucidate these issues long-term].

This study's results support further trials on this topic in the Cardio-Oncology area. Also, it should be kept in mind that the potential utilization of FFR-guided PCI should be based on careful clinicians’ decisions to apply FFR, according to the individual medical context of each patient.

The Authors may consider some suggestions - for instance, the Authors can:

P # 1 TITLE

a suggestion:

Fractional flow reserve (FFR) - should be used rather that ‘FFR’ only

P # 7 Discussion

The strengths and limitations of the study should be pointed out [added to the discussion].

 Thank you

Round 2

Reviewer 1 Report

Thank you for authors' contribution. The reviewer's concerns were resolved.